# Influence of the Microbial Metabolite Acetyl Phosphate on Mitochondrial Functions Under Conditions of Exogenous Acetylation and Alkalization

**DOI:** 10.3390/metabo14120703

**Published:** 2024-12-13

**Authors:** Natalia V. Beloborodova, Nadezhda I. Fedotcheva

**Affiliations:** 1Federal Research and Clinical Center of Intensive Care Medicine and Rehabilitology, Petrovka St., 25-2, Moscow 107031, Russia; nvbeloborodova@yandex.ru; 2Institute of Theoretical and Experimental Biophysics, Russian Academy of Sciences, Institutskaya St., 3, Pushchino 142290, Russia

**Keywords:** acetyl phosphate, alkalization, nonenzymatic acetylation, respiration, adenine nucleotide translocase, mitochondrial permeability transition pore, succinate dehydrogenase

## Abstract

Background. Acetyl phosphate (AcP) is a microbial intermediate involved in the central bacterial metabolism. In bacteria, it also functions as a donor of acetyl and phosphoryl groups in the nonenzymatic protein acetylation and signal transduction. In host, AcP was detected as an intermediate of the pyruvate dehydrogenase complex, and its appearance in the blood was considered as an indication of mitochondrial breakdown. In vitro experiments showed that AcP is a powerful agent of nonenzymatic acetylation of proteins. The influence of AcP on isolated mitochondria has not been previously studied. Methods. In this work, we tested the influence of AcP on the opening of the mitochondrial permeability transition pore (mPTP), respiration, and succinate dehydrogenase (SDH) activity under neutral and alkaline conditions stimulating the nonenzymatic acetylation using polarographic, cation-selective, and spectrophotometric methods. Results. It was found that AcP slowed down the opening of the mPTP by calcium ions and decreased the efficiency of oxidative phosphorylation and the activity of SDH. These effects were observed only at neutral pH, whereas alkaline pH by itself caused a decrease in these functions to a much greater extent than AcP. AcP at a concentration of 0.5–1 mM decreased the respiratory control and the swelling rate by 20–30%, while alkalization decreased them twofold, thereby masking the effect of AcP. Presumably, the acetylation of adenine nucleotide translocase involved in both the opening of mPTP and oxidative phosphorylation underlies these changes. The intermediate electron carrier phenazine methosulfate (PMS), removing SDH inhibition at the ubiquinone-binding site, strongly activated SDH under alkaline conditions and, partially, in the presence of AcP. It can be assumed that AcP weakly inhibits the oxidation of succinate, while alkalization slows down the electron transfer from the substrate to the acceptor. Conclusions. The results show that both AcP and alkalization, by promoting nonmetabolic and nonenzymatic acetylation from the outside, retard mitochondrial functions.

## 1. Introduction

The human microbiome produces functional small molecules regulating health and disease. Acetyl phosphate (AcP) is a microbial metabolite involved in the acetate kinase/phosphate acetyltransferase pathway of central bacterial metabolism. This pathway interconverts coenzyme A, ATP, and acetate by two-step reactions to provide the reversible synthesis of AcP or ATP, depending on bacterial growth conditions [1]. The cellular concentrations of acetate, AcP, and CoA determine which reactions are predominant under particular physiological conditions [2]. As was shown, the intracellular concentration of AcP in wild-type *Escherichia coli* cells reaches 3 mM, and it can function as a donor of both acetyl and phosphoryl groups [3]. As a rule, acetylation by AcP in bacteria occurs nonenzymatically and correlates strongly with the fluctuating levels of AcP. The investigation of the role of AcP-mediated nonenzymatic acetylation in *E. coli* showed its participation in the inhibition of enzymes of the central bacterial metabolism. in particular, of the glycolysis and the tricarboxylic acid cycle [4,5]. Moreover, nonenzymatic acetylation by AcP determines, among other effects, the survival of bacteria in the host via the modulation of chemotaxis, virulence, and antibiotic resistance [6,7]. As a donor of phosphoryl groups, AcP can participate in signal transduction [3,8,9] and ADP phosphorylation [10].

In mammalian mitochondria, AcP was detected as an intermediate of the pyruvate dehydrogenase complex, which forms and rapidly degrades during the oxidation of pyruvate [11]. A high level of AcP in the human blood was found in pathologies of spinal cord injury and was interpreted as an indication of myopathy and mitochondrial breakdown [12]. Among various human blood metabolites whose appearance indicates metabolic disturbances in cancer, the presence of AcP was found to significantly correlate with a decrease in the risk of lung cancer [13]. In vitro experiments showed that AcP serves as a powerful acetylating agent. The kinetic analysis of nonenzymatic protein acetylation revealed a high dependence of the reaction rate on the concentration of AcP. In tests of the acetylation of glutamate dehydrogenase and other mitochondrial proteins by AcP and acetyl-CoA, the reaction rate was found to increase by an order of magnitude with an increase in their concentration from 1 to 8 mM, with a certain advantage for AcP [14]. As was found, enzyme-independent acetylation prefers an alkaline pH and a high abundance of the acetylating agent to drive the reaction. Therefore, mitochondria possessing these properties are considered to be the main location for nonenzymatic acetylation [15,16,17]. Using the proteomic analysis, it was shown that 63% of proteins localized in mitochondria contain acetylated amino acid residues [18,19]. A hyperacetylation of mitochondrial proteins and enzymes was identified in a number of pathologies, including neurodegenerative diseases, cancer, obesity, diabetes, and others [20].

Regarding the enzymes of the tricarboxylic acid cycle, it was found that succinate dehydrogenase (SDH), malate dehydrogenase, and fumarate hydratase contain the greatest number of acetylated sites [21]. A large number of acetylation sites in mitochondria compared to other compartments is related to the fact that mitochondria contain a high concentration of acetyl-CoA, a donor of acetyl groups. Its concentration in the matrix can reach 2 mM [22]. Nonenzymatic acetylation can also occur in mitochondria along with enzymatic acetylation carried out by acetyltransferases [15,17].

It is assumed that acetylation changes the properties of proteins such as activity, conformation, protein–protein interactions, stabilization, and subcellular localization. These parameters are important in the regulation of the mitochondrial permeability transition pore (mPTP). The formation of the mPTP is mediated by several participants, including adenine nucleotide translocase (ANT), ATP synthase, a phosphate carrier, and the outer-membrane voltage-dependent anion channel [23,24]. Some data show that the acetylation of cyclophilin D and ANT influences their conformation, binding, and mPTP opening. As was found, the increased acetylation of CyPD following myocardial ischemia-reperfusion facilitated mPTP opening [25]. ANT acetylation decreased the apparent affinity of ADP for ANT binding [26,27].

We previously showed that AcP diminished the activity of cyclooxygenase when incubated with the pure enzyme, but had little effect on the enzyme activity when incubated with monocyte cell lysate, presumably due to the spreading of the effect to other targets [28]. Also, our investigation of the influence of microbial metabolites on the nonspecific permeability of mitochondria revealed a small protective effect of AcP on the Ca^2+^-induced mPTP opening [29]. We hypothesize that AcP, as a microbial metabolite, may be an effective marker of severe or latent infections. Since the only biochemical property of AcP is the donation of acetyl groups for non-enzymatic acetylation processes, we tested its influence on mitochondrial functions in this context. In the present work, the influence of AcP on respiration, dehydrogenase activity, and mPTP opening in isolated rat liver mitochondria under neutral and alkaline conditions was investigated to assess the possible contribution of non-metabolic exogenous acetylation from outside to mitochondrial processes.

## 2. Materials and Methods

All reagents (lithium potassium acetyl phosphate, 2,6-dichlorophenolindophenol, phenazine methosulfate, adenosine diphosphate, and others) were from the Sigma-Aldrich Corporation (St. Louis, MO, USA).

Isolation of rat liver mitochondria. The study was conducted in accordance with the ethical principles formulated in the Helsinki Declaration on the care and use of laboratory animals. Manipulations were carried out by the certified staff of the Animal Department of the Institute of Theoretical and Experimental Biophysics (Russian Academy of Sciences) and approved by the Commission on Biomedical Ethics of ITEB RAS (N1/2024, 18 March 2024). Experiments were performed on male rats of the Wistar line. Liver mitochondria were isolated using a standard method of differential centrifugation in a medium containing 300 mM sucrose, 1 mM EGTA, and 10 mM Tris-HCl buffer (pH 7.4). Samples of mitochondria were washed with the isolation medium without EGTA, suspended in a medium of the same composition, and stored on ice [29,30].

Determination of mitochondrial respiration. Oxygen consumption in a mitochondrial suspension was determined by the polarographic method with a Clark-type electrode in a closed chamber of 2 mL containing 2.0 mg of mitochondrial protein under continuous stirring. Mitochondrial respiration was supported by succinate (5 mM). Respiration was activated by ADP (200 µM) for the evaluation of coupled respiration and by carbonyl cyanide 4-(trifluoromethoxy)phenylhydrazone (FCCP, 1 µM) for the evaluation of uncoupled respiration. The respiratory control index was defined as the ratio of the ADP-stimulated respiration rate to the respiration rate after ADP phosphorylation as described earlier [30]. AcP at concentrations of 100–1000 μM was added to the cuvette with mitochondria and substrate; incubation continued for three minutes before the addition of ADP. These experimental conditions were applied at neutral (7.2) and alkaline (8.0) pH.

Determination of the activity of succinate dehydrogenase by dichlorophenolindophenol reduction. The activity of succinate dehydrogenase was determined by the reduction in an electron acceptor, dichlorophenolindophenol (DCPIP) [31,32]. Mitochondria (0.5 mg protein/mL) were incubated in 2 mL of medium containing 125 mM KCl and 15 mM HEPES, pH 7.2, in the presence of 1 mM cyanide, 20 μL of 10% Triton X-100, and 100 μM DCPIP. The DCPIP reduction reaction was induced by the addition of 5 mM succinate, activated with 100 μM PMS, and the recovery rate of the acceptor was measured at a wavelength of 600 nm using an Ocean Optics USB4000 spectrophotometer (Ocean Optics, Dunedin, FL, USA). AcP at concentrations of 0.5–1 mM was added to the cuvette with all reagents except succinate and PMS and incubated during three minutes before the addition of succinate. Such experimental conditions were applied at neutral (7.2) and alkaline (8.0) pH.

Determination of the activity of succinate dehydrogenase was performed by the reduction in 3-(4,5-dimethylthiazole-2-Il)-2,5-diphenyl-tetrazolium bromide (MTT). Mitochondria (0.5 mg protein/mL) were added to 2 mL of an incubation medium containing 125 mM KCl, 15 mM HEPES, pH 7.2, 150 μM MTT, and 1 mM succinate and incubated for 5 min [32,33]. Preincubation of the mitochondria with AcP at concentration 1 mM was carried out in the absence of the substrate for 5 min; succinate and MTT were then added and incubated for an additional 5 min. After the incubation, 20 μL of 10% Triton X-100 was added to each sample to complete mitochondrial lysis, and the optical density of the reduced electron acceptor was measured at 580 nm using an Ocean Optics USB4000 spectrophotometer (Ocean Optics, Dunedin, FL, USA). These experimental conditions were used at neutral (7.2) and alkaline (8.0) pH.

Measurement of the membrane potential, swelling rate, and calcium retention capacity of mitochondria. The difference in electric potentials on the inner mitochondrial membrane was determined by the distribution of a lipophilic cation tetraphenylphosphonium (TPP^+^), whose concentration in the medium was recorded with a tetraphenylphosphonium-selective electrode. The accumulation of Ca^2+^ ions in the mitochondria was recorded with a Ca^2+^(-)selective electrode as the change in calcium concentration in the external medium in response to successive CaCl_2_ additives at a final concentration of 20 μM with a “Record 4” device (Russia) and computer recording. The calcium retention capacity of mitochondria was determined by the ability of mitochondria to accumulate and to retain successive additions of calcium ions at the threshold concentration necessary for the opening of the nonspecific mitochondrial pore as described earlier [29,30]. The incubation medium contained 125 mM KCl, 1.5 mM KH2PO4, and 15 mM HEPES (pH 7.25). The swelling of mitochondria was measured at a wavelength of 540 nm using an Ocean Optics USB4000 spectrophotometer (Ocean Optic, Dunedin, FL, USA). The swelling was induced by CaCl_2_ addition and was assessed by the changes in optical density. Mitochondria at a concentration of protein of 0.3–0.4 mg/mL were incubated in the buffer containing 125 mM KCl, 15 mM HEPES, 1.5 mM phosphate, and 5 mM substrate [29,30]. AcP in different concentrations (from 100 to 1000 μM) was added to the cuvette with mitochondria and substrate, incubation continued for three minutes before the addition of CaCl_2._ These experimental conditions were used at neutral (7.2) and alkaline (8.0) pH.

Experimental conditions for testing the effect of AcP on mitochondrial functions. In each experiment, AcP in the concentration range from 100 to 1000 μM was added to the cuvette with mitochondria and the corresponding reagents during the recording of each indicator of the function; the incubation of AcP with mitochondria continued for 3–5 min before adding the inducer of the functional parameter being tested. These experimental conditions were used both at neutral (7.2) and alkaline (8.0) pH. Additional experimental conditions are given in the captions to the figures. The figures show the data of typical experiments performed in at least three replicates with different samples of mitochondria.

Statistical analysis. The data given represent the means ± standard error of means (SEM) from five to seven experiments, or are the typical traces of three to five identical experiments with the use of different mitochondrial samples. Statistical significance was estimated by the Student’s *t*-test with *p* < 0.05.

## 3. Results

### 3.1. Influence of AcP on Mitochondrial Respiration Under Neutral and Alkaline Conditions

Figure 1 shows the influence of AcP on mitochondrial respiration in neutral and alkaline incubation mediums. At neutral pH (7.2), AcP in the concentration range from 100 μM to 1 mM affected predominantly ADP-stimulated respiration and did not almost influence the respiratory rate at rest and in the presence of FCCP (Figure 1a). With the addition of ADP, AcP decreased the ADP/O ratio and oxidative phosphorylation due to a longer phosphorylation time and an incomplete restoration of the respiratory rate after ADP phosphorylation. Taking into account that non-enzymatic acetylation is activated by alkalization, we compared the effect of AcP on respiration at neutral and alkaline pH.

It turned out that alkalization (pH 8.0) significantly altered all respiration parameters compared to the neutral pH control. As shown in Figure 1b, at alkaline pH, the respiration rates are significantly decreased, by almost 50%, in all states (insert). The ADP-stimulated respiration changed to the greatest extent, which showed up in a strong decrease in the ADP/O ratio (more oxygen consumed for ADP phosphorylation) and the respiratory control, defined as the ratio of the ADP-stimulated respiration rate to the respiration rate after ADP phosphorylation. AcP, which has the same direction of action, did not exert any influence under these conditions and did not enhance the alkalization effect (Figure 1c). The overall effect of AcP and alkalization is a decrease in the respiratory control by 30% and 50%, respectively (Figure 1d). Changes in the duration of ADP phosphorylation were also observed when assessing the effects of AcP and alkalization on the decrease/recovery of the membrane potential in response to ADP supplementation. In these experiments, the decelerating effect of AcP was greater at alkaline pH than at neutral pH (Figure 1e). In this case, the deceleration was 25% of the control at this pH, while the alkalization itself slowed down the process by 35–40% (Figure 1f).

### 3.2. Influence of AcP on Succinate Dehydrogenase Activity Under Neutral and Alkaline Conditions

To assess the contribution of possible inhibition of succinate dehydrogenase (SDH) by AcP and alkalization to changes in respiration, we investigated their effect on enzyme activity using DCPIP as an electron acceptor. At neutral pH, AcP (1 mM) had a noticeable inhibitory effect on the enzyme activity, decreasing the rate of DCPIP reduction during succinate oxidation (Figure 2a). The intermediate electron carrier PMS activated the reaction, but the rate of DCPIP reduction did not reach the control level. Alkalization itself greatly decreased the reaction rate, and the addition of AcP (1 mM) did not enhance this inhibition (Figure 2b). However, PMS did not completely remove the inhibition in the presence of AcP compared to the control. As shown in the insert in Figure 2b, PMS did not abolish the inhibition of DCPIP reduction by malonate, an inhibitor of the catalytic subunit of SDH, but readily eliminated the inhibition by TTFA, an inhibitor of the ubiquinone-binding subunit. Consequently, it can be assumed that AcP weakly inhibits the oxidation of succinate, while alkalization slows down the electron transfer from the substrate to the acceptor. As shown in Figure 2c, AcP and alkalization decreased the rate of DCPIP reduction by 25–30% and 45–50%, respectively. When MTT was used as an electron acceptor, AcP decreased its reduction only after the incubation of mitochondria with AcP in the absence of substrate; in this case, inhibition reached 40% (Figure 2d).

### 3.3. Influence of AcP on the Mitochondrial Permeability Transition Pore Under Neutral and Alkaline Conditions

The examination of the influence of AcP on the mitochondrial permeability transition pore (mPTP) is of particular interest, since in this case it can act on some components of the pore from the outside, which is important in the context of the microbial origin of AcP. Figure 3 shows the influence of AcP on mPTP opening, as measured by changes in the threshold calcium concentrations required to induce the pore and in resistance to calcium loading. At neutral pH, AcP at a concentration of 0.5–1.0 mM weakly increased the threshold calcium concentrations required for pore opening and slowed down the release of accumulated calcium from mitochondria (Figure 3a). At alkaline pH, the threshold calcium concentration increased almost twofold (Figure 3b). Under these conditions, AcP did not have a noticeable effect on the calcium retention capacity (Figure 3c). Thus, AcP has only a weak effect on the calcium retention capacity at neutral pH and no effect at alkaline pH (Figure 3d).

Figure 4 shows the influence of AcP on Ca^2+^-induced pore opening as measured by mitochondrial swelling. In these experiments, the rate of swelling induced by a threshold calcium concentration was assessed. As shown in Figure 4a, at neutral pH, AcP slowed down the swelling rate compared to the control.

This effect was observed at an AcP concentration of 0.5–1 mM. Alkalization led to a strong extension of the lag phase and a twofold decrease in the swelling rate (Figure 4b). Under these conditions, AcP either had no effect or caused a slight activation, within 10–20%, of the swelling rate (Figure 4c). Thus, AcP has a moderate protective effect on the opening of mPTP at neutral pH and has no noticeable effect at alkaline pH, which itself sharply decreases the swelling rate (Figure 4d).

## 4. Discussion

Our results show that AcP, a microbial metabolite and a strong acetylating compound, influences the opening of mPTP, oxidative phosphorylation, and SDH activity. This study is the first to test the effects of AcP on intact isolated mitochondria. In both bacteria and mitochondria, AcP is formed during the oxidation of pyruvate. In bacteria, AcP is synthesized enzymatically from acetyl-CoA, a product of the pyruvate dehydrogenase reaction, and phosphate [1,2], while in mitochondria it is formed as an intermediate during the oxidation of pyruvate by pyruvate dehydrogenase [11] and its accumulation is considered as an indication of mitochondrial breakdown [12]. AcP as a microbial metabolite can be involved in the development of mitochondrial dysfunctions underlying the multiple organ failure in infections and sepsis. Thus, the hyperacetylation of mitochondrial proteins and enzymes has been identified in a number of pathologies, including sepsis, inflammation, neurodegeneration, and other diseases [34,35,36]. Metabolic hyperacetylation is mediated mainly by excessive production of acetyl-CoA and deficiency of NAD, as well as by lowered expression of mitochondrial deacetylases, as it was found in the mouse heart in conditions of LPS-induced sepsis [34]. In this case, the level of acetylated enzymes of the TCA cycle, including citrate synthase, aconitase, succinate dehydrogenase catalytic subunit A, fumarate hydratase, and malate dehydrogenase was significantly increased.

Considering the predominantly microbial origin of AcP, we evaluated its influence on isolated mitochondria as a possible factor in their dysfunction in bacteremia and sepsis. As follows from our data, AcP can have both a positive protective effect and a toxic effect on mitochondria. The protective effect includes an increase in resistance to calcium load and a decrease in the rate of mitochondrial swelling during the mPTP opening compared to the control. As known, the mPTP participates in the regulation of both physiological and pathological processes, providing the low conductance associated with the redistribution of ions and small metabolites in physiological conditions and the high conductance that promotes the efflux of large molecules, which can lead to cell death [23,24]. Consequently, this effect of AcP can be attributed to protective effects that promote cell survival. The toxic effects of AcP are associated with a decrease in oxidative phosphorylation and SDH activity. According to our data, AcP in the concentration range of 0.5–1 mM mainly changed ADP-stimulated respiration, decreasing the respiratory control index.

Regarding the effect of acetylation on the mPTP opening, the data are variable. It has been shown that the hyperacetylation of the mitochondrial protein stimulates permeability transition pore opening in rats with obesity [37]. This study revealed the key role of acetylation of CypD in the increase in the sensitivity of the mPTP to Ca^2+^ overload, which was reflected in an about 30% decrease in the Ca^2+^ retention capacity. Also, the binding of CypD to the inner membrane components of the pore was positively related to its acetylation state [25]. On the other hand, it was also shown that it is free CypD that is needed to target the mPTP, while any CypD modification, including acetylation, desensitizes mPTP opening [38,39]. It should be noted that variations in the influence of acetylation on mPTP may be explained by the fact that different amino acid residues (lysine, cysteine, and others) undergo acetylation. In our previous studies, the incubation of isolated mitochondria with N-acetylimidazole, which acetylates predominantly tyrosine residues, resulted in an activation of mPTP opening by calcium ions [32].

Adenine nucleotide translocase (ANT) is another component of mPTP that undergoes acetylation with changes in activity. ANT exchanges cytosolic adenosine diphosphate for mitochondrial adenosine triphosphate through the inner mitochondrial membrane. Also, ANT is considered as the main component of the mitochondrial pore that interacts with other pore-forming proteins, including cyclophilin D, to form the mPTP. As was shown, the acetylation of ANT decreased ADP affinity and the adenine nucleotide flux as well as altered oxidative phosphorylation [40]. Based on these data, it can be assumed that, in our experiments, AcP could influence the activity of mPTP and oxidative phosphorylation due to the acetylation of ANT as a participant in both these processes. It should be noted that all the above-mentioned studies were performed on cells and biopsies in which the level and the effect of acetylation were assessed by the number of acetylated sites and changes in the function during the overexpression of deacetylase, respectively. Our approach is based on the extramitochondrial localization of AcP as a microbial metabolite so that the direct influence of the acetylating agent on the function can be assessed. Despite these distinctions, the effects of acetylation on the quantitatively examined functions do not significantly differ, not exceeding 30% variations in both experimental conditions.

Our data show that SDH is most susceptible to the influence of AcP, which is consistent with the data on the preferential inhibition of this enzyme by metabolic acetylation [16,18]. In our experiments, the inhibition of SDH by AcP was observed during the incubation of lysed mitochondria with DCPIP as an acceptor and was not evident when measuring the respiration in intact mitochondria. These distinctions may be explained either by the availability of AcP to the enzyme or by the competitive inhibition with the substrate. The latter assumption is supported by the data with MTT as an acceptor, which showed that the strongest inhibition of SDH occurs upon preincubation of mitochondria with AcP in the absence of the substrate. In this case, the inhibition of SDH reached 40%. Incomplete recovery of the reaction rate in the presence of PMS, an intermediate electron carrier, also indicates the inhibition of the catalytic subunit of SDH, since PMS completely abolished only the inhibition induced by TTFA, an inhibitor of the ubiquinone-binding site, without affecting the malonate-induced inhibition.

Moreover, PMS reversed the inhibition of SDH caused by alkalization. Generally, alkalization led to the same changes as incubation with AcP; namely, it increased the calcium retention capacity and decreased oxidative phosphorylation. However, the changes induced by alkalization were much stronger, masking the expected increase in the acetylation reaction. In our experiments, the influence of AcP was still observed under alkalization conditions, since PMS is unable to remove the inhibition of the catalytic subunit and, as a consequence, activate the reduction in the acceptor, DCPIP, to the control level. Indeed, PMS, acting as an intermediate carrier of electrons from the source to the acceptor, makes it possible to localize the site of inhibition.

Previously, we noted a protective influence of alkalization against the mPTP opening in the experiments with iron and pro-oxidant loading [41] as well as a positive influence on the enzyme activity associated with an inhibition of the ubiquinone-binding site by redox-active microbial metabolites [32]. In this regard, the study of the relationship between these mitochondrial functions under alkaline conditions is of particular interest. It can be assumed that several factors are involved in the protective effect of alkalization against mPTP opening. The most obvious is the protection against both acidification and the activation of lipid peroxidation during calcium loading. Another factor is related to a slowdown in the rate of respiration and electron transfer from the enzyme to the oxygen as an acceptor. However, this does not limit the resulting increase in the accumulation of calcium ions. The question of whether these factors are related to each other or are independent requires further clarification.

The possibility for AcP as a microbial metabolite to reside in the host is supported by the data on the detection of other acetylated microbial compounds in the blood. Recently, the appearance of acetylated polyamines of bacterial origin in the blood was found during bloodstream infections in induced sepsis [42]. Moreover, these microbial acetylated polyamines, especially acetyl putrescine, were suggested to be used as markers of infection [42]. It is interesting that, earlier, we applied polyamines to protect mitochondrial enzymes against acetylation. According to our data, spermidine decreased the acetylation of mitochondrial enzymes during the incubation with the acetylating agent N-acetylimidazole, which was explained by the involvement of acetyl groups in the acetylation of polyamine [33]. As is known, the acetylation of polyamines is a key reaction of their catabolism since their intracellular concentration is controlled by spermidine acetyltransferase that forms the excreted metabolite diacetylspermidine [43,44]. Apparently, polyamines can prevent the hyperacetylation of enzymes during the excessive accumulation of acetylating compounds.

The data obtained show that AcP has a moderate effect on two mitochondrial functions, the opening of mPTP, and oxidative phosphorylation. Since acetylation is the main property of this compound, it can be assumed that the acetylation of ANT, a participant in both functions, underlies the observed changes. The moderate inhibition of SDH observed in experiments with artificial acceptors is consistent with the data on the preferential acetylation of this enzyme in vivo, also with a moderate loss of activity. According to our data, the greatest inhibition by AcP occurs in the absence of a substrate, which indicates a competitive character of inhibition.

As shown recently, the sepsis-induced myocardial dysfunction was accompanied by a decreased expression of deacetylases and hyperacetylation of the enzymes of the tricarboxylic acid cycle SDH, fumarate hydratase, and malate dehydrogenase as well as the accumulation of succinate and oxaloacetate compared to control values [34]. A decrease in oxidative phosphorylation was documented in an experimental study on a mouse sepsis model [45]. Twelve hours after the start of the experiment, a significant decrease in oxidative phosphorylation was recorded under the influence of LPS, as well as a significant increase in anaerobic metabolism in hepatocytes. Through the use of metabolomic analysis, a metabolic shift from a carbohydrate-based metabolism to a metabolism using amino acids and fatty acids has been confirmed; this was accompanied by an increase in the number of intermediates and derivatives of the tricarboxylic acid cycle [45]. Also, our previous study revealed a close correlation between the elevated levels of succinic acid and microbial phenolic acids in the blood of critically ill patients at different stages of sepsis [46]. Thus, our studies on the involvement of microbial metabolites in the regulation of mitochondrial functions show that, along with redox-active aromatic metabolites having both protective and toxic properties, AcP can also influence mitochondrial processes due to its acetylating activity.

The literature data from proteomic analysis show that the acetylation of mitochondrial enzymes often does not affect their activity since numerous amino acid residues, including those not associated with the active center of the enzyme, are subjected to acetylation [17,18]. We identified two functional targets of AcP in mitochondria, SDH and mPTP. Potential sites of SDH acetylation by AcP were revealed using their selective inhibitors and electron acceptors. The identification of AcP targets among mPTP components, the most probable of which is ANT, requires further investigation.

We applied alkalization with the purpose of enhancing the effect of acetylation. Presumably, the binding of acetyl groups is accompanied by the deprotonation of the amino acid residues of a protein, and subsequently, by the acidification of the medium, which is also known to inhibit the reaction. Therefore, alkalization should activate the acetylation process. Moreover, alkalization is applied to restrict excess acidosis in severe infections and sepsis [47,48]. As our data show, alkalization slows down the mPTP opening, which is physiologically significant in terms of cell survival. Also, AcP and alkalization act on the tested mitochondrial functions (respiration, mPTP opening, SDH activity) in the same direction, namely towards their inhibition. This effect can have either a positive or negative meaning, depending on the physiological or pathological conditions.

In pathogenesis, under aerobic conditions, the production of AcP by bacteria is activated due to the involvement of pyruvate oxidase, which is important for the bacterial aerobic growth and trafficking across host barriers [49]. It is important to emphasize that the analysis of causal relationships in the development of mitochondrial dysfunction in sepsis indicates the leading role of microbial metabolism. In particular, this statement is confirmed by the deepest taxonomic and metabolic disorders in the microbiota, which not only accompany, but in most cases precede the development of sepsis [50]. It has been shown that, normally (in healthy people with a “healthy” microbiota), the products of microbial metabolism of aromatic amino acids entering the systemic circulation are the final metabolites, while microbial metabolites associated with sepsis are intermediate. An interesting analogy is that AcP is also a microbial intermediate. In addition, according to published data [12], the concentration of AcP in the blood may be quite high in patients with severe diseases, which may indicate an excessive intake of AcP from the intestine due to incomplete metabolic pathways of the anaerobic microbiota.

## 5. Conclusions

The origin of AcP can be either microbial, associated with bacterial overgrowth and/or impaired microbiota metabolism, or mitochondrial, associated with impaired metabolism/degradation of mitochondria, or both phenomena simultaneously. In any case, its appearance in the blood indicates the presence of a pathological process. In the near future, it is necessary to evaluate the pathogenic role, diagnostic and prognostic significance of AcP, and the expediency of the monitoring of this metabolite, especially in critically ill patients.

## Figures and Tables

**Figure 1 metabolites-14-00703-f001:**
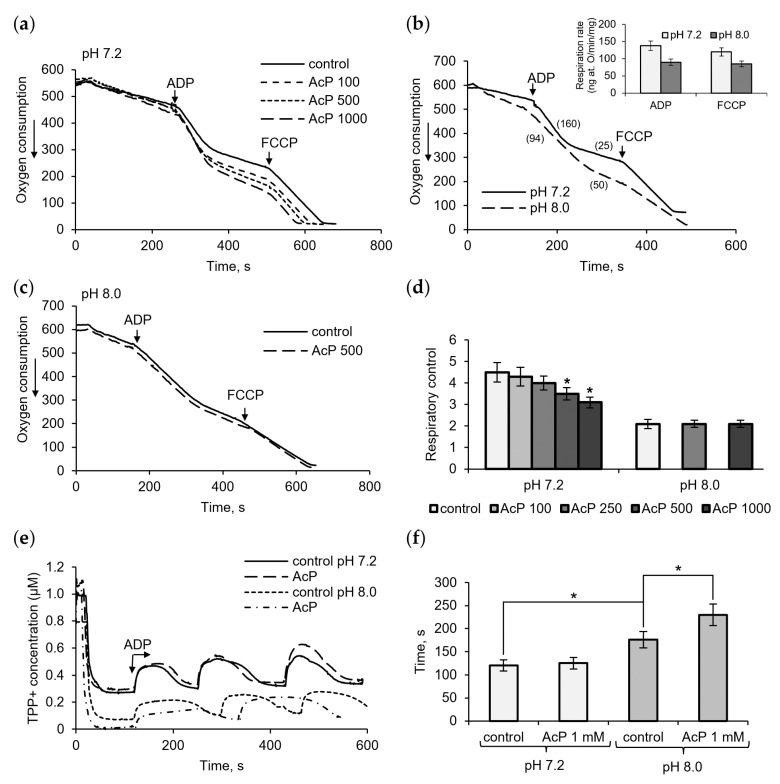
Influence of AcP on respiration and oxidative phosphorylation. The effect of AcP on ADP- and FCCP-activated mitochondrial respiration at neutral (**a**) and alkaline pH (**b**,**c**) and on the respiratory control index at indicated concentration of AcP (**d**); influence of AcP (1 mM) on the membrane potential under successive additions of ADP, 50 μM each (**e**), and the time of phosphorylation (**f**) at neutral and alkaline pH. Respiration rates are indicated in parentheses; asterisks indicate the values that differ significantly from the control values (*p* < 0.05).

**Figure 2 metabolites-14-00703-f002:**
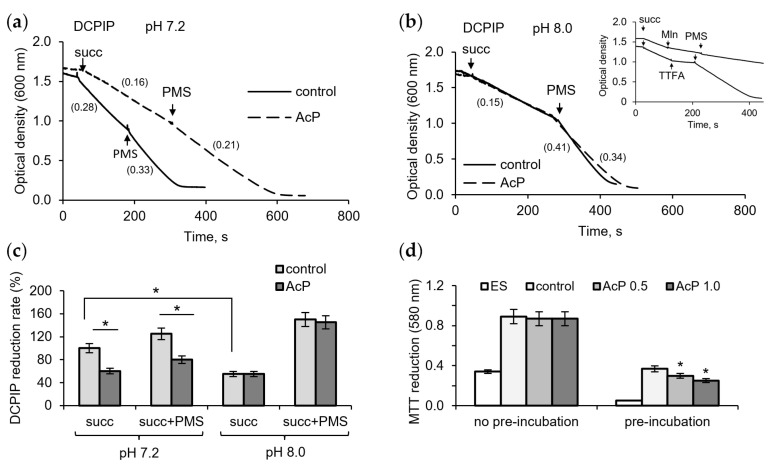
Influence of AcP on succinate dehydrogenase activity under neutral and alkaline conditions. The influence of AcP (1 mM) on DCPIP reduction at neutral (**a**) and alkaline (**b**) pH; the action of PMS (100 µM) on the malonate (Mln, 1 mM)- and TTFA (50 µM)-induced inhibition (**b**, insert); influence of AcP (1 mM) and PMS on DCPIP reduction rate (**c**) and MTT reduction (**d**) under neutral and alkaline pH. Reaction rates are indicated in parentheses; asterisks indicate the values that differ significantly from the control values (*p* < 0.05).

**Figure 3 metabolites-14-00703-f003:**
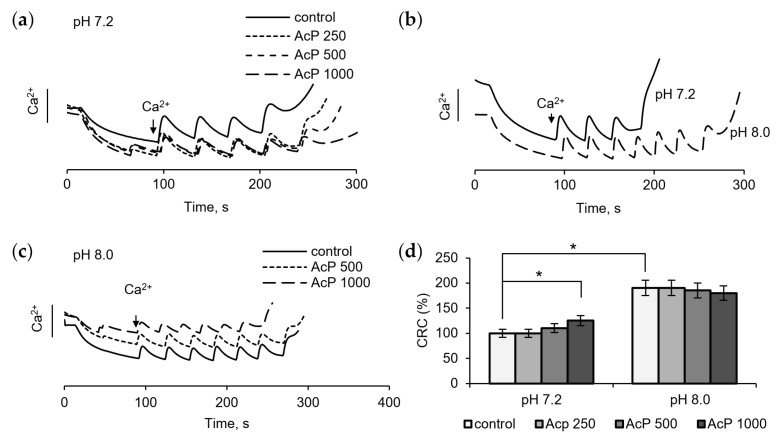
Influence of AcP on the threshold calcium concentrations inducing mPTP opening under neutral and alkaline conditions. Influence of AcP at indicated concentrations on the calcium retention capacity at neutral pH under successive additions of CaCl_2_, 25 μM each (**a**); the calcium retention capacity at neutral and alkaline pH (**b**); influence of AcP at indicated concentrations on the calcium retention capacity at alkaline pH (**c**); influence of AcP at indicated concentrations on the threshold calcium concentrations inducing mPTP opening under neutral and alkaline conditions (**d**). Asterisks indicate the values that differ significantly from the control values (*p* < 0.05).

**Figure 4 metabolites-14-00703-f004:**
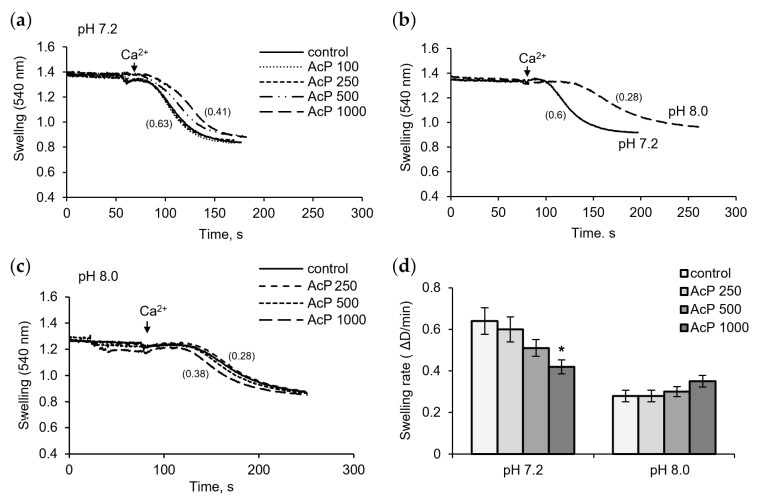
Influence of AcP on the mitochondrial swelling induced by calcium under neutral and alkaline conditions. Influence of AcP at indicated concentrations on the swelling of mitochondria induced by CaCl2 (50 M) at neutral pH (**a**); swelling rates at neutral and alkaline pH (**b**); influence of AcP at indicated concentrations on the swelling at alkaline pH (**c**); swelling rates at different concentrations of AcP at neutral and alkaline pH (**d**). Swelling rates are indicated in parentheses; asterisk indicates values that differ significantly from the control value (*p* < 0.05).

## Data Availability

The datasets generated during and/or analyzed during the current study are available from the corresponding author upon reasonable request.

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
