# Peer review of "Influence of the Microbial Metabolite Acetyl Phosphate on Mitochondrial Functions Under Conditions of Exogenous Acetylation and Alkalization"

_metabolites, 2024, doi:10.3390/metabo14120703_

Round 1

Reviewer 1 Report

Comments and Suggestions for Authors

To improve the manuscript author suggested to address the following comments

SDH abbreviations should be used where it reported at first.

Line 27 - What is PMS?

Line 48 - “.” missing after reference [4,5].

What is the reason for using male animal for the present study?

Line 117 - repetition of “of” 2 times.

Each section of the methodology needs to be addressed with reference.

Author needs to provide the company details for USB4000 spectrophotometer.

Line 164 Figure 1A - needs to be written as Figure 1a.

Author needs to follow figure marks as same in the writing part. Eg. Figure 1 a, 1 b…. or 1 A, 1 B….

Author Response

Replay 1

Thank you very much for reviewing our work and helpful comments. We have made the necessary corrections indicated by you.

To improve the manuscript author suggested to address the following comments

 SDH abbreviations should be used where it reported at first.

The error is corrected.

Line 27 - What is PMS?

We added the chemical name PMS and gave an abbreviation: phenazine methosulfate (PMS)

Line 48 - “.” missing after reference [4,5].

The error is corrected.

What is the reason for using male animal for the present study?

It is generally accepted to use male rats as they are physiologically more stable.

Line 117 - repetition of “of” 2 times.

The error is corrected.

Each section of the methodology needs to be addressed with reference.

We have added references in each experimental section.

Author needs to provide the company details for USB4000 spectrophotometer.

We have added details - Ocean Optic, Dunedin, FL, USA

Line 164 Figure 1A - needs to be written as Figure 1a.

Author needs to follow figure marks as same in the writing part. Eg. Figure 1 a, 1 b…. or 1 A, 1 B….

We have corrected this error everywhere in the text.

Reviewer 2 Report

Comments and Suggestions for Authors

Purpose of this study is not understandable. Authors examined effect of AcP on mPTP, respiration, and SDH under neutral and alkaline conditions stimulating the nonenzymatic acetylation in isolated mitochondria. I understand AcP is involved in the central bacterial metabolism, and eukaryote mitochondria contains AcP. I guess authors expect acetylation of proteins by AcP observed in bacteria is also occur in eukaryotic mitochondria that contains AcP. However, the experiments in this study are not linked with protein acetylation, nor the enzymatic activities except for SDH. Authors tested effect of AcP on SDH, mPTP, and respiration in artificial conditions of alkaline pH, but the experimental and physiological meaning is not persuading. I do not agree to significance of the test of AcP effect on mitochondria in this study.

Comments on the Quality of English Language

There are several grammatical errors in the manuscript, which needs to be revised.

Author Response

Replay 2.

Purpose of this study is not understandable. Authors examined effect of AcP on mPTP, respiration, and SDH under neutral and alkaline conditions stimulating the nonenzymatic acetylation in isolated mitochondria. I understand AcP is involved in the central bacterial metabolism, and eukaryote mitochondria contains AcP. I guess authors expect acetylation of proteins by AcP observed in bacteria is also occur in eukaryotic mitochondria that contains AcP. However, the experiments in this study are not linked with protein acetylation, nor the enzymatic activities except for SDH. Authors tested effect of AcP on SDH, mPTP, and respiration in artificial conditions of alkaline pH, but the experimental and physiological meaning is not persuading. I do not agree to significance of the test of AcP effect on mitochondria in this study.

Thank you very much for reviewing our work and for the important questions you raised. We have taken these aspects into account in our studies, and now we hope to clarify them in our responses to you and additions to the manuscript.

Purpose of this study is not understandable. Authors examined effect of AcP on mPTP, respiration, and SDH under neutral and alkaline conditions stimulating the nonenzymatic acetylation in isolated mitochondria. I understand AcP is involved in the central bacterial metabolism, and eukaryote mitochondria contains AcP. I guess authors expect acetylation of proteins by AcP observed in bacteria is also occur in eukaryotic mitochondria that contains AcP.

The purpose of our work was to test the possible contribution of AcP as a microbial metabolite to the mitochondrial dysfunction, a key factor in the development of multiple organ failure in infections and sepsis. Since the only biochemical property of AcP is the donation of acetyl groups to non-enzymatic acetylation processes, we tested its effect on mitochondrial functions in this context. In addition, our study is the first to examine the effects of AcP on intact isolated mitochondria. We assume that the exogenous non-metabolic acetylation by AcP can occur in vivo since AcP was found in the blood in severe pathologies and, moreover, its aerobic production by bacteria increased significantly during bloodstream infections. Also, we hypothesize that AcP, as a microbial metabolite, may be an effective marker of severe or latent infections and we plan a research on this topic in our clinical studies.

To clarify the purpose of our study, we have made the following addition at the end of the Introduction:

We hypothesize that AcP, as a microbial metabolite, may be an effective marker of severe or latent infections. Since the only biochemical property of AcP is the donation of acetyl groups for non-enzymatic acetylation processes, we tested its influence on mitochondrial functions in this context.

However, the experiments in this study are not linked with protein acetylation, nor the enzymatic activities except for SDH.

We took into account these aspects based on the proteomic approach to the analysis of metabolic acetylation in cells and mitochondria. The literature data on proteomic analysis show that the acetylation of mitochondrial enzymes often does not affect their activity since numerous amino acid residues, including those not associated with the active center of the enzyme, are subjected to acetylation. We identified two functional targets of AcP in mitochondria, SDH and mPTP. Potential sites of SDH acetylation were revealed using their selective inhibitors and electronic acceptors. We plan to determine the influence of AcP on the regulation of mPTP opening also using numerous selective inhibitors and protectors of adenine nucleotide translocase, a key component of the pore. Also, we will search for an opportunity for proteomic analysis if AcP will be detected in the blood in our clinical studies.

We have made the following addition at the end of the Discussion:

The literature data from proteomic analysis show that the acetylation of mitochondrial enzymes often does not affect their activity since numerous amino acid residues, including those not associated with the active center of the enzyme, are subjected to acetylation [17,18]. We identified two functional targets of AcP in mitochondria, SDH and mPTP. Potential sites of SDH acetylation by AcP were revealed using their selective inhibitors and electron acceptors. The identification of AcP targets among mPTP components, the most probable of which is ANT, requires further investigation.

Authors tested effect of AcP on SDH, mPTP, and respiration in artificial conditions of alkaline pH, but the experimental and physiological meaning is not persuading. I do not agree to significance of the test of AcP effect on mitochondria in this study.

We tested the influence of AcP on mitochondria both in neutral and alkaline pH.  We applied alkalization with the sole purpose of enhancing the effect of acetylation. According to the literature data, the binding of acetyl groups is accompanied by the deprotonation of the amino acid residues of a protein, and consequently by the acidification of the medium, which is also known to inhibit the reaction. Therefore, alkalization should activate the acetylation process. As for the physiological significance, alkalization is applied to reduce excess acidosis in serious infections and sepsis. As our data show, alkalization inhibits the mPTP opening, which is also physiologically significant in terms of cell survival. In our experiments, AcP and alkalization act on the tested mitochondrial functions (respiration, mPTP opening, SDH activity) in the same direction, namely, towards inhibition. This effect can have either a positive or negative meaning, depending on the physiological or pathological conditions.

We have made the following addition at the Discussion

We applied alkalization with the purpose to enhance the effect of acetylation. According to the literature data, the binding of acetyl groups is accompanied by the deprotonation of the amino acid residues of a protein, and subsequently, by the acidification of the medium, which is also known to inhibit the reaction. Therefore, alkalization should activate the acetylation process. Moreover, alkalization is applied to restrict excess acidosis in severe infections and sepsis []. As our data show, alkalization slows down the mPTP opening, which is physiologically significant in terms of cell survival. Also, AcP and alkalization act on the tested mitochondrial functions (respiration, mPTP opening, SDH activity) in the same direction, namely, towards their inhibition. This effect can have either a positive or negative meaning, depending on the physiological or pathological conditions.

There are several grammatical errors in the manuscript, which needs to be revised.

Our institute professional translator carefully checked our manuscript. Errors and typos were found and corrected.

Thank you again for your comments. We think that the additions to the text you have initiated will significantly improve the presentation and understanding of our work.

Reviewer 3 Report

Comments and Suggestions for Authors

The present manuscript titled “Influence of the Microbial Metabolite Acetyl Phosphate on Mitochondrial Functions under Conditions of Exogenous Acetylation and Alkalization” led by Natalia shows the novel findings on the role of AcP on mitochondria. The findings are novel, and the manuscript is well-written. However, the following issues must be resolved by the authors.

·       Section 2: The subheads of this section should be in bold.

·       Provide a reference for the OCR method.

·       The following is missing in the methodology: Authors have treated mitochondria with AcP and then investigated mitochondria function. However, the concentration of AcP and duration of AcP incubation is missing in the methodology.

·       Authors have carried out the experiments under different pH conditions, but these details are missing in the methodology.

·       Further add a separate subheading experimental section which should clearly state the duration and concentrations of AcP.

·       According to the title AcP influenced mitochondrial function through acetylation and alkalization. However, authors have not proved the acetylation in mitochondrial subunits. Though authors have given literature support for the acetylation property of AcP, validating it in this study would be more advantageous.

Author Response

Replay 3

Thank you very much for reviewing our work and helpful comments. We have made the necessary corrections indicated by you.

The present manuscript titled “Influence of the Microbial Metabolite Acetyl Phosphate on Mitochondrial Functions under Conditions of Exogenous Acetylation and Alkalization” led by Natalia shows the novel findings on the role of AcP on mitochondria. The findings are novel, and the manuscript is well-written. However, the following issues must be resolved by the authors.

  • Section 2: The subheads of this section should be in bold.

We have highlighted the section titles in bold.

  • Provide a reference for the OCR method.

We have added a reference to this method.

  • The following is missing in the methodology: Authors have treated mitochondria with AcP and then investigated mitochondria function. However, the concentration of AcP and duration of AcP incubation is missing in the methodology.

We have added the description of methods:

AcP at concentrations of 100 - 1000 μM was added to the cuvette with mitochondria and the substrate; the incubation continued for three minutes before the addition of ADP. These experimental conditions were applied at neutral (7.2) and alkaline (8.0) pH.

  • Authors have carried out the experiments under different pH conditions, but these details are missing in the methodology.

We added the phrase in each section that the same experimental conditions were used at neutral (7.2) and alkaline (8.0) pH.

  • Further add a separate subheading experimental section which should clearly state the duration and concentrations of AcP.

We added a separate section about this:

Experimental conditions for testing the effect of AcP on mitochondrial functions. In each experiment, AcP in the concentration range from 100 to 1000 μM was added to the cuvette with mitochondria and the corresponding reagents during the recording of each indicator of the fuction; the incubation of AcP with mitochondria continued for 3-5 min before adding the inducer of the functional parameter being tested. These experimental conditions were used both at neutral (7.2) and alkaline (8.0) pH. Additional experimental conditions are given in the captions to the figures. The figures show the data of typical experiments performed in at least three replicates with different samples of mitochondria.

  • According to the title AcP influenced mitochondrial function through acetylation and alkalization. However, authors have not proved the acetylation in mitochondrial subunits. Though authors have given literature support for the acetylation property of AcP, validating it in this study would be more advantageous.

According to your suggestion, we have made the additions to the manuscript. In the Introduction section, we have added:

We hypothesize that AcP, as a microbial metabolite, may be an effective marker of severe or latent infections. Since the only biochemical property of AcP is the donation of acetyl groups for non-enzymatic acetylation processes, we tested its influence on mitochondrial functions in this context.

In the Discussion section, we have added:

The literature data from proteomic analysis show that the acetylation of mitochondrial enzymes often does not affect their activity since numerous amino acid residues, including those not associated with the active center of the enzyme, are subjected to acetylation [17,18]. We identified two functional targets of AcP in mitochondria, SDH and mPTP. Potential sites of SDH acetylation by AcP were revealed using their selective inhibitors and electron acceptors. The identification of AcP targets among mPTP components, the most probable of which is ANT, requires further investigation.

We applied alkalization with the purpose to enhance the effect of acetylation. Presumably, the binding of acetyl groups is accompanied by the deprotonation of the amino acid residues of a protein, and subsequently, by the acidification of the medium, which is also known to inhibit the reaction. Therefore, alkalization should activate the acetylation process. Moreover, alkalization is applied to restrict excess acidosis in severe infections and sepsis [47, 48]. As our data show, alkalization slows down the mPTP opening, which is physiologically significant in terms of cell survival. Also, AcP and alkalization act on the tested mitochondrial functions (respiration, mPTP opening, SDH activity) in the same direction, namely, towards their inhibition. This effect can have either a positive or negative meaning, depending on the physiological or pathological conditions.

Round 2

Reviewer 2 Report

Comments and Suggestions for Authors

With this version of the manuscript, this study is acceptable for publication.

Reviewer 3 Report

Comments and Suggestions for Authors

Authors have addressed all the issues satisfactorily